# Hierarchical effects facilitate spreading processes on synthetic and empirical multilayer networks

**Casey Doyle***, **Thushara Gunda, Asmeret Naugle**

Sandia National Laboratories, Albuquerque, NM, United States of America

* cldoyle@sandia.gov

**Data Availability Statement:** All relevant data are within the paper and its Supporting information files.

**Funding:** Sandia National Laboratories is a multimission laboratory managed and operated by

## Abstract

In this paper we consider the effects of corporate hierarchies on innovation spread across multilayer networks, modeled by an elaborated SIR framework. We show that the addition of management layers can significantly improve spreading processes on both random geometric graphs and empirical corporate networks. Additionally, we show that utilizing a more centralized working relationship network rather than a strict administrative network further increases overall innovation reach. In fact, this more centralized structure in conjunction with management layers is essential to both reaching a plurality of nodes and creating a stable adopted community in the long time horizon. Further, we show that the selection of seed nodes affects the final stability of the adopted community, and while the most influential nodes often produce the highest peak adoption, this is not always the case. In some circumstances, seeding nodes near but not in the highest positions in the graph produces larger peak adoption and more stable long-time adoption.

## Introduction

The diffusion and spread of new practices is a crucial part of the evolution of societies and organizations, particularly in regards to the adoption innovations or procedural updates. Due to a litany of structural and cultural barriers, however, innovation spread can be slow and eventually lead to the death of the new innovation. This situation creates a stagnant environment that stifles production and growth; thus identifying and removing these barriers to adoption is crucial for the efficiency and health of long-standing organizations. Complicating any study on this type of structure, however, are the various channels that communications can flow through, often including formal hierarchical networks as well as informal social and working connections. In this paper, we focus on the differences between these formal and informal structures and how innovation adoption is affected by the characteristics of each. Additionally, we look at selection strategies for the seed nodes, or innovators within the system, and compare the costs and benefits of having highly influential nodes as the first adopters in the system.

National Technology & Engineering Solutions of Sandia, LLC, a wholly owned subsidiary of Honeywell International Inc., for the U.S. Department of Energy's National Nuclear Security Administration under contract DE-NA0003525. This paper describes objective technical results and analysis. Any subjective views or opinions that might be expressed in the paper do not necessarily represent the views of the U.S. Department of Energy or the United States Government. The Sandia National Laboratories website is https://www.sandia.gov/ The funders helped procure the data for this work.

**Competing interests:** This study was funded by Sandia National Laboratories, a multimission laboratory managed and operated by National Technology & Engineering Solutions of Sandia, LLC, a wholly owned subsidiary of Honeywell International Inc., for the U.S. Department of Energy's National Nuclear Security Administration under contract DE-NA0003525. This does not alter our adherence to PLOS ONE policies on sharing data and materials, and the data used in this study is all contained within the supplementary materials of the paper.

The basic mechanism of innovations adoption is captured by the Dosage Model [1], an elaboration on the classic epidemiological susceptible-infected-recovered (SIR) [2] model that is extended to contain more detail and capture the memory aspects of social contagion. As an agent based model with memory effects, this framework is particularly well-suited to studying how small changes in the dynamics of state transitions or structural differences in the system affect the overall success of a new idea. This functionality is critical to studying innovation diffusion, as the significant role of peer communication [3] and the influence of clustering [4] are essential to properly modeling the adoption processes. Further, the application of organizational change provides a strong incentive to study the minimal commitment or seed size that still leads to wide-scale adoption on the network, a hallmark of epidemiological models that focus on identifying critical nodes in the epidemic process [5, 6].

Using epidemiological models as a base system for social simulations in this way has become a common practice, and the concept of social contagion has proven to be a useful tool in modeling how information and even innovation travels in complex networks [7–10]. Still, while the spreading processes themselves have striking similarities, the two applications differ greatly in the end goal of the simulation; epidemiological models are generally concerned with identifying the nodes that can be immunized or the structures that resist spreading whereas social systems are often concerned with nodes and structures that *facilitate* spread. Further, the varied and overlapping forms of possible communication between nodes opens the door for different scales and methods of transmission, leading naturally to the study of multilayered and interconnected networks. These underlying structures can better capture the effects of differing types of communications, such as leadership communications, that affect the final decision making process of individuals [10].

Due to their high degree of applicability to real world systems, multilayered networks and their effects on classical spreading models have become a field of great interest in recent years [11–13], in particular for the ability of the interconnected structure to allow for epidemics under parameterizations that would stifle spreading on a monoplex network [14–16]. This popularity has led to studies of different types of structures, rules, and metrics to measure the effects of the multilayered structures, with topics ranging from network resiliency to lexical, negotiation, and opinion spread applications [17–21]. In particular, efforts to find the optimal immunization strategies using SIR dynamics in multilayered networks have shown differences in optimal immunization strategies depending on the underlying network and distribution of immunizations across layers [22–24]. Further studies on the topic have shown epidemics that are unable to cross layers on networks lacking sufficient interconnectedness [25], and even effects on spreading in the case of competing diseases and procedurally-generated networks [26, 27]. Finally, other dynamic systems including the threshold model [28] and prisoner's dilemma [21] have also found optimal strategies for multilayered networks, and the problem has even been solved via mean-field considerations [29].

The theoretical groundwork laid by these prior studies has paved the way for more detailed considerations of the topic using empirical data to inform the network structures. In this paper we take this approach, building a system that considers real-world effects in two ways; first, we use the Dosage Model rather than a pure SIR consideration, allowing us to tailor the dynamics to our system of focus. In particular, this paper is interested in innovation and procedural change in large, multi-level organizations, and as such the memory aspect of the dosage model is useful for creating the resistance towards change characteristic of individuals in corporate hierarchies. Second, we define our network structure in terms of a management hierarchy, first building synthetic networks that mirror what can be found in the real world, taking data from the Sandia National Laboratories (SNL) organizational structure to build two competing networks that represent different modes of social interaction and possible spread.

This allows us to tailor our model to a specific real-world process and compare different communication channels and seed selection strategies to determine the optimal choices for sustained adoption.

## Methods

### Multilayer dosage model

The dynamics of the model discussed here are based on the Dosage Model proposed by Dodds and Watts [1], incorporating the same three states as the SIR model (susceptible, infected, and removed) with a more complicated state transition scheme (see Fig 1). Rather than flat probabilities controlling the state transitions, nodes undergo transitions based on how many *doses* towards switching they have received in a given memory window, $T$. Each node $i$ has an individual dosage threshold, $d_i^*$, drawn from a Gaussian distribution with a mean of 1, a standard deviation of 0.5, and a minimum $d_i^* > 0$, that controls their eligibility for state transitions and allows for a heterogeneous reluctance towards switching throughout the population. Each micro-time step, defined as one interaction, a node is selected to be active in the role of 'listener'. A subsequent node is selected from that node's connections to be the 'speaker'. If the 'speaker' is in the infected ($I$) state, they confer a random dose, drawn from a Gaussian distribution with a mean of $0.5\langle d^*\rangle$ and a standard deviation of $0.25\langle d^*\rangle$, to the 'listening' node. If the 'speaker' is not in state $I$ then they simply confer a dose of 0. Each node keeps a memory record of its last $T$ doses (for this paper, we use a memory window $T = 5$ to set a characteristic time for our simulations), and upon receiving a new dose (including null doses), 'listening' nodes check their cumulative dose at that time, $D_{t,i} = \sum_{t_0=t-T}^{t_0=t} d_{t_0}$ (where $d_{t_0}$ is the dosage the node received during transaction $t_0$) against their dosage threshold. If they are in state $S$ and $D_{t,i} > d_i^*$, they transition to state $I$. Conversely, if they are in state $I$ and $D_{t,i} < d_i^*$, they have a probability $r = 1/T$, or $r = 0.2$, of entering the removed state ($R$), signifying that they have abandoned the new innovation due to perceived lack of support. This removal rate is chosen for its symmetry with the infection memory time scale, however a brief investigation of varying removal rates is also conducted. In our formulation state $R$ is absorbing, with no probability of removed nodes to become infected again, though they remain active nodes in the simulation capable of conferring doses of 0 to other nodes. As such, the only possible ending state of the system in the long time limit is the eradication of the $I$ state as that is the only system state with no further possible state changes. This is a common feature of *SIR* models, of which the model used here is an elaboration on and thus follows similar trajectories. An important difference, though, is that by utilizing the memory window $T > 1$ the system obtains the ability to enter metastable states that provide long tails on surviving active communities [1], the effect of

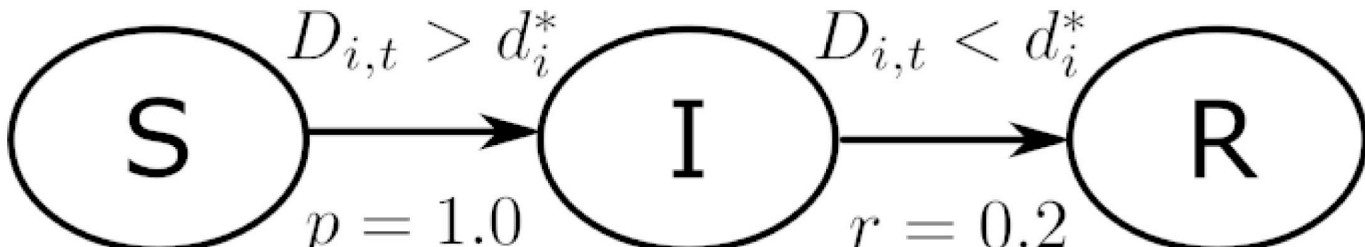

**Fig 1. Dosage model state transitions.** Each node has three possible states, *S*, *I*, and *R*. Nodes gain doses towards switching states every time they 'listen' to another node; doses are nonzero if the 'speaking' node is in state *I* and zero otherwise. When a node in state *S* has a cumulative dosage over the past *T* time steps, $D_{i,t}$, greater than its threshold, $d_i^*$, it switches to state *I*. When a node in state *I* has a cumulative less than its threshold, it has a 20% change per activation to switch to state *R*. State *R* is absorbing.

which is analyzed here. In this paper we utilize only a single value for $T$ with limited testing on varying values of $r$ as the sensitivity of these values has been studied previously, showing that increasing the size of the memory window increases the tail end of the simulation and persistence of the $I$ state exponentially. By holding this value constant in our simulations we can investigate other values that affect the peak spread and persistence of the infection, revealing structures that favor these metastable states. Further discussion of the state behavior over time as well as plots of the state transitions and prior work on parameter sensitivity is included in S1 Appendix.

We further modify the basic version of the dosage model above to fit the scenario of corporate or organizational change by adding in multi-layer considerations to the dynamics. The original dosage model is already well-suited to the memory and peer-influence heavy, slow moving nature of this sort of large scale organization of interest, but the addition of management layers requires extra consideration for the differences in peer communications versus communications from supervisors. To this end, we define multilayer communication rules such that intra-layer communications (e.g., staff to staff) and upward inter-layer communications (e.g., staff to manager) are defined as outlined above. Downward inter-layer communications (e.g., manager to staff), however, have greater weight if the manager is pushing adoption. To accomplish this, if a node from a higher layer in the graph is in the $I$ state, when they speak to a node from a lower layer they always confer a dose equal to the downstream node's threshold. This guarantees an immediate switch for the downstream node, but leaves them vulnerable to permanent removal if they were not already familiar with the adopted idea or do not have their adoption reinforced by others within $T$ time steps, fitting with organizational mandates that can easily lose staff interest if not supported widely.

The selection of this model is motivated by the application of organizational innovation spread, with each of the compartments and state transitions being chosen for those purposes. In this work, the $S$ state represents nodes that are unaffiliated with and have not adopted the new innovation or procedure. The $I$ compartment represents adopters that are actively promoting the innovation, and the $R$ compartment represents the rejection of the innovation and its promotion. The R state is made absorbing in the model, meaning that nodes cannot leave that state, to account for the disenchantment and active resistance that is seen in innovation adoption [30], as well as to reflect the specific context of management and organizational innovation rejection, where there are many common causes and perspectives to explain lasting innovation rejection [31]. The focus of this model is on the social and hierarchical pressures on the adoption of the innovation, and as such the infection has no internal drive for individuals to remain in that state. In application, this means that the modeled innovation would not have a positive efficiency that would allow it to be self-sustaining or to resurge once rejected. Since our study is on the effects of these social pressures, using a neutral framework allows us to examine only those effects on the system. Additionally, our time scale and static view of the innovation does not allow for long term effects such as the evolution of the innovation, further indicating the likelihood of long term rejection. Finally, there is no state transition directly from $S$ to $R$ under the assumption of good faith on the part of the agents. Nodes in state $I$ and nodes in state $R$ behave identically in their outward interactions, both passing on zero dose in their interactions, but only nodes in state R that have negative firsthand experience with the innovation present the active resistance to deny a large social pressure coming either from a superior node or multiple peers.

For each model run, seeding is handled randomly by layer, testing the spreading due to a minimal seed (1 node) and that of the introduction of an established community (0.015$N$ nodes). Nodes are randomly selected and are all within the layer of interest for that test, i.e.,

staff seeding will place either 1 or $0.015N$ nodes in state $I$ randomly throughout the staff layer but will track infection spread across all layers.

## Synthetic networks

First, as a baseline for our investigation, we create synthetic versions of the multilayer manager networks to understand the primary effects and effect sizes expected of the dosage model with advantaged managerial layers. To this end, we start with a base layer built from a random geometric graph (RGG) [32], chosen for its community oriented structure, with $N_{staff} = 1000$ nodes, $dim = 2$, $p = 2$, and $r = (\langle k \rangle / (N_{staff}^* \pi))^{1/2}$, where the average degree $\langle k \rangle = 20$ to match empirical data and $N_{staff}$ is the size of the base layer of the hierarchy. This graph is then partitioned into $c$ different communities, where $c = .06N_{staff}$ or 60 communities in this case to match the management representation at the first level of the empirical networks. Communities are partitioned using the Fluid Communities algorithm [33], chosen for its ability to assign a consistent number of communities across many runs. Doing so fully divides the system, leading to a skewed normal distribution of community sizes with an average community size of 16.667 nodes, a standard deviation of 5.642, and with skewing parameter $\alpha = 2.6$, location $\xi = 10.417$ and scale $\omega = 8.420$. Then a second layer for the management nodes is spawned as an independent network before being interconnected to the staff network with interlayer edges connecting each manager node to the staff nodes in a single downstream community. The number of nodes in the management layer is exactly equal to the number of partitions created in the staff layer, $m = c = 0.06N$. This is exactly analogous to the interlayer connections used in the empirical datasets, while the intralayer connections between the managers are created independently based on various random graph models including no connections, full connections, RGG, Erdős–Rényi (ER) random graphs [34], and Barabási–Albert random graphs [35] with the same $\langle k \rangle = 20$ as well as ER graphs of varying $\langle k \rangle$. For these networks, $\langle k \rangle = 20$ was chosen to match the empirical networks described in the next section. Further explanation of the community size distribution and synthetic network creation are captured in S1 Appendix and S1 Sample code respectively.

## Empirical networks

After the analysis of various synthetic network structures, we study the spreading effects on an empirical hierarchical dataset based on the employment structure of Sandia National Laboratories. Access to this data was overseen by the Sandia National Laboratories data governance committee, and all data was fully anonymized before being made available to the authors. From this data we build two separate graphs, one that contains the strict corporate organization lines at the time of study, referenced here as the Line-Org graph, and one that leverages the time individuals spent working together over a two year period prior to draw connections between nodes, referenced here as the Project/Task (Project) graph. Both networks use the same population pool and thus have the same number of layers, $N = 1942$ and $n_l = 5$ respectively. The first layer has $N_1 = 1787$ nodes in it, while the second has $N_2 = 115$ nodes, third has $N_3 = 33$ nodes, fourth $N_4 = 6$ nodes, and the final layer has just a single node $N_5 = 1$. The data to reconstruct these networks are included as Supporting Files. Additionally, S1 Appendix contains a replication of the key analysis done here on an existing publicly available manufacturing dataset for comparison [36, 37].

**Line-Org graph.** The first empirical graph used contains the straight organizational structure based on official department listings and job titles. For intra-layer connections, this means that each staff member is connected to everyone else in their department and have no connections outside of it (i.e., a staff member is department 1111 will be connected only to all

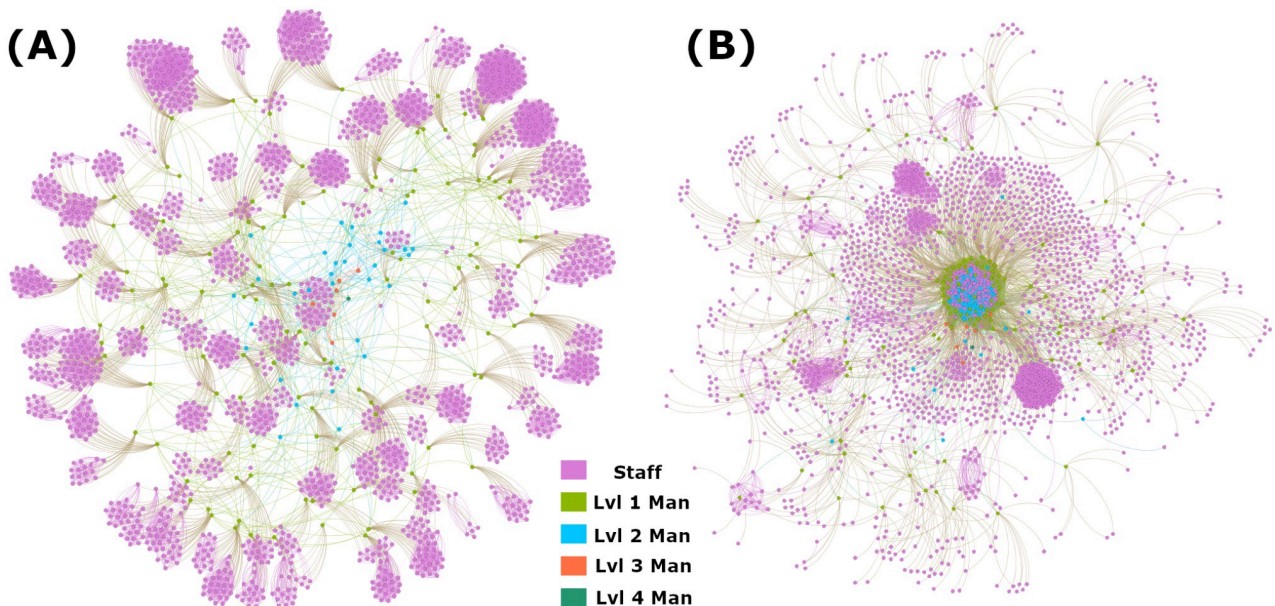

**Fig 2. Empirical network visualizations.** Nodes are color coded by layer: staff nodes are pink, level 1 managers are green, level 2 managers are light blue, level 3 managers are orange, and level 4 is dark blue. A: Visualization of the Line-Org network. B: Visualization of the Project network.

other staff members in department 1111). Similarly, a manager will be connected to all other managers within their group on the second layer (i.e., a manager for department 1111 will be connected only to other managers for groups 111$X$). This continues up the chain until all 5 layers in the dataset are accounted for. The inter-layer connections are controlled via the management lines, for example the manager of 1111 is connected downstream to all staff in department 1111 and upstream to the second level manager of group 111$X$. In this way, the network is controlled against isolated nodes through the single connecting director at the top of the chain in level 5. As shown in Fig 2(A) and Table 1, this network is highly modular with each individual department being very insular and relying on the upper layers to connect groups to each other. It has extreme clustering coefficients due to the high number of triangles within the intra-layer connections and low closeness centrality due to the lack of connection between groups. Finally, it is characterized by a fairly consistent degree distribution across layers as the groups are designed to be reasonably similar in size.

**Table 1. Network statistics of empirical networks by layer.**

| | Line-Org | | | Project | | |
|---|---|---|---|---|---|---|
| | $\langle C_{centrality} \rangle$ | $\langle k \rangle$ | $\langle C_{clustering} \rangle$ | $\langle C_{centrality} \rangle$ | $\langle k \rangle$ | $\langle C_{clustering} \rangle$ |
| Layer 1 | 0.16 | 20.05 | 0.99 | 0.27 | 13.75 | 0.29 |
| Layer 2 | 0.19 | 20.85 | 0.60 | 0.35 | 93.21 | 0.46 |
| Layer 3 | 0.22 | 9.73 | 0.57 | 0.36 | 100.09 | 0.64 |
| Layer 4 | 0.27 | 12.33 | 0.48 | 0.31 | 11.33 | 0.39 |
| Layer 5 | 0.26 | 7.0 | 1.0 | 0.26 | 9.0 | 0.58 |
| Total | 0.16 | 19.89 | 0.97 | 0.28 | 19.91 | 0.31 |

Shows various centrality and clustering statistics for each empirical network to inform on the structure, where $\langle C_{centrality} \rangle$ is the average closeness centrality, $\langle k \rangle$ is the average degree, and $\langle C_{clustering} \rangle$ is the average clustering coefficient for the nodes in each layer.

**Project graph.** As mentioned, the second empirical graph uses the same nodes as the first, but alters the intra-layer connections such that they are defined by time spent concurrently on a project rather than strict corporate structure. This way, this graph represents a more social network that considers how interactions may occur organically between regular work confluence rather than through mandates by defined organizations within the company. To accomplish this, we draw edges between any two individuals that have worked on projects together for more than 370 hours total (regardless of how many projects the time is split between); the threshold of 370 hours was chosen to preserve the overall average degree from the Line-Org network. The inter-layer connections remain the same, as manager communication to staff maintains a level of corporate formality that is well represented by the direct line from superior to subordinate. In comparison to the Line-Org network, Fig 2 and Table 1 show that the Project network is highly centralized with the first two management levels serving as the major hub tying the majority of nodes together. It has higher closeness centralities across the board, while the degree distribution skews heavily towards the management layers as they are generally the nodes that have worked most broadly and for the longest periods of time within the network. In general, the clustering coefficient is significantly lower than the Line-Org network except in the dense management layers where clustering coefficients are comparable.

## Results and discussion

Using the above network bases and spreading processes, we model innovation adoption in large scale hierarchical systems. In particular, we look at what structures and parameterizations lead to the largest general adoption as well as the most robust population against removal and subsequent death of the innovation. Simulations are run for $1, 000, 000$ micro-timesteps ($\sim 1000$ epochs where each epoch is defined as $N$ interactions) and averaged over 1000 runs.

### Synthetic networks

For the synthetic network outlined above, we first study the baseline case of the dosage model on a RGG with no management layer yielding a maximum concurrent adoption of $I_{max} = 0.027$ with only a single seed node and $I_{max} = 0.141$ with $0.015N$ seed nodes. Comparing these numbers to those where a management layer is included in the system, shown in Table 2, a significant increase in spreading with the multilayer structure is evident. Even without connections between the managers, the max adoption using only a single staff node doubles, and with a larger seed shows a 68% increase. Further, the performance of staff-seeded simulations is notably worse than using manager nodes as the seed, where over $\sim 10\%$ of the population is reachable with only a single seed and to nearly half of the total population with a seed size of

**Table 2. Maximum adoption for different manager connection strategies.**

|  | Staff Seed | | Manager Seed | |
|---|---|---|---|---|
|  | **Minimal Seed** | **Seed 0.015N** | **Minimal Seed** | **Seed 0.015N** |
| Isolated | 0.056 | 0.237 | 0.107 | 0.346 |
| Fully Connected | 0.095 | 0.354 | 0.265 | 0.550 |
| RGG, $\langle k \rangle = 20$ | 0.098 | 0.357 | 0.241 | 0.472 |
| BA, $\langle k \rangle = 20$ | 0.101 | 0.359 | 0.228 | 0.473 |
| ER, $\langle k \rangle = 20$ | 0.102 | 0.355 | 0.230 | 0.470 |

The average maximum adoption over 1000 runs on the synthetic network with various manager connection strategies. Average degrees of 20 were chosen for the management layer to match the base layer and the overall average of the empirical networks.

$0.015N$. These trends are further emphasized when connections are created between the management nodes, in particular showing significant increases in the case of low seed sizes, where adoption in the system nearly doubles again with the addition of connections between managers. When managers are the seed, this further increase is even greater, and while more modest in the cases of large seeds the effect is still pronounced.

Interestingly, the structure of the connections between managers tends to matter less than the presence of the connections in the first place, as there are not significant differences between the three random graph structures tested, and further relatively minor differences exist between the fully connected manager layer and the randomly connected management layers. Table 2 shows only a small benefit towards the fully connected graph over the random graphs when the seed is on the management layer, while there is actually a very slight suppressive effect on the staff seeded nodes. This small suppression is minor enough to be within the error of the simulations, but may also reflect the management nodes' ability to insulate themselves from the downstream nodes when they are highly connected. In the synthetic structure used here, management nodes are connected to an average of 16.67 downstream nodes, so while a certain level of connection between them allows for a more effective leveraging of the management spreading advantage, a high level of connection quickly attaches them to other mangers more so than downstream nodes. With a fully connected management layer, this means there are 3.6 times more management connections than there are staff connections, and when the seed is entirely contained in the staff layer this can diminish the speed with which the innovation moves across layers. On the contrary, this problem becomes moot if the seed is contained on the management layer, and the advantages of spreading on the higher network layer manifest more. In fact, as shown in Fig 3(A), with an ER connected management layer the maximum adoption increases for low connection densities between the management nodes and then levels off after a probability of connection of $\sim 0.3$, corresponding to the point at which the average inter-layer degree of the management layer ($\langle k \rangle = 18$) is greater than the average intra-layer degree. The exception to this is the case of a large management seed, where increased connection of managers can no longer hinder the spread for that layer and thus only supports the overall adoption on the network. Finally, we look at the stability of the spread

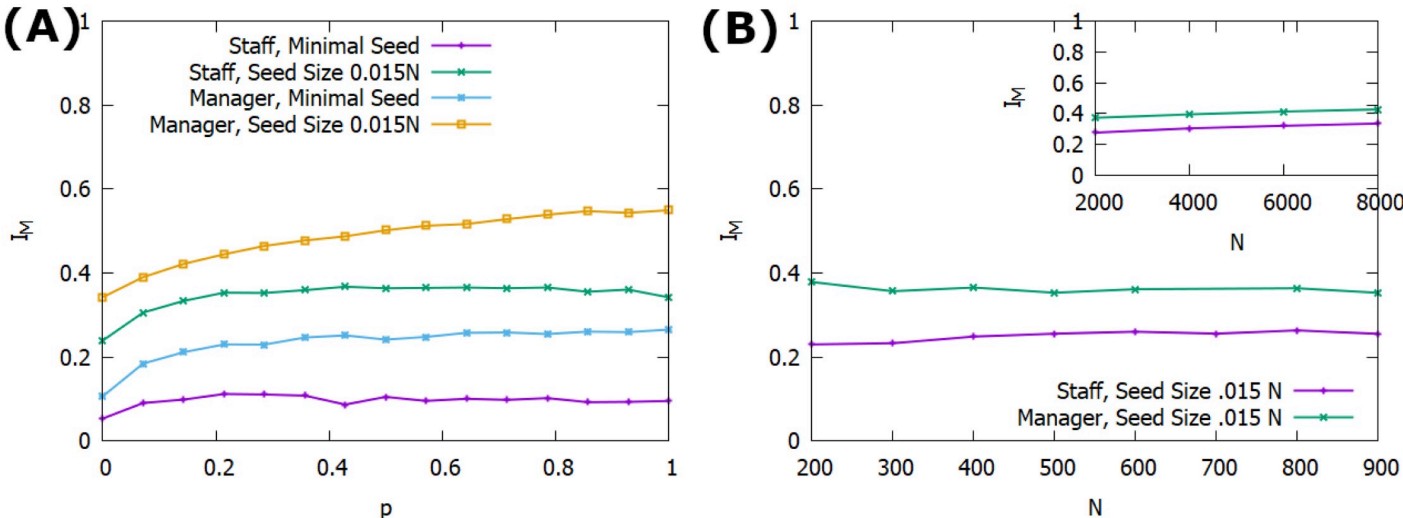

**Fig 3. Synthetic network spread.** Plot of average max adoption over 1000 runs. $I_M$ is the maximum adoption, $p$ is the probability of connection for every pair of managers, and $N$ is the overall system size. Management connections are made via ER random graph method. A: Average maximum adoption versus management connection density over 1000 runs. B: Average maximum adoption vs sysem size. Seed size is proportional total network size, set at 1.5% of the network. All networks have a number of managment communities $c = 0.06N$ and an average node degree of $\langle k \rangle = 20$.

across system sizes, where Fig 3(B) shows that the average maximum infected population is stable for systems both smaller and larger than the $N = 1000$ case used in prior analysis. This analysis is done using a seed size of 1.5%, matching the seed size in in both the preceding analysis and that performed on the empirical networks. Additionally, the number of managers and the communities they represent is modulated for the system size, $c = 0.06N$ and the average degree across all layers of the network is constant at $\langle k \rangle = 20$ by keeping manager connections minimal.

## Empirical networks

**SI dynamics.** To begin the analysis of multilayered dosage spreading on empirical networks, we first look at a simplified form with a removal rate of 0. This modification effectively makes the dynamics that of an SI system where the *I* state is absorbing, so rather than evaluating based on maximum adoption we consider the time taken to reach consensus where all nodes are in the *I* state. The results of this basic consideration can be see in Fig 4, showing that for all cases except those with weak seeding, the Project network outperforms the Line-Org network. When the seed is small and not highly influential ($< 5$ staff or $< 2$ level 1 managers), the Project network is liable to get stuck in metastable states and take extreme times to reach consensus, likely due to the higher probability of the seeding being placed in a remote, near-isolated node in this network. Additionally, the advantages of the Project network are most apparent in the later portions of the simulation as the system reaches high levels of consensus, indicating that the spreading in this type of network has faster convergence with a shorter tail despite the presence of near-isolated nodes. This is likely due to the fact that spreading is faster to near-isolated *nodes* (characteristic of the Project network) than to near-isolated *clusters* (characteristic of the Line-Org netowrk), as there is no group consensus to combat. If a node is attached only to one other node, then it is sufficient to wait for the Poisson selection process to activate that node after its sole neighbor has switched states, whereas a near-isolated community can self-perpetuate for long periods of time by becoming akin to an echo chamber. Finally Fig 4 shows that outside of extremely small seeds, there is virtually no difference between the

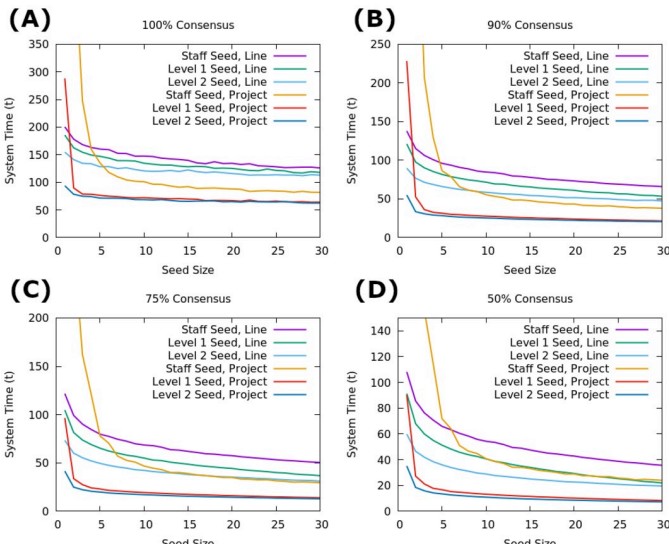

**Fig 4. Time to consensus in SI framework.** Time to different levels of consensus with an absorbing *I* state for varying seed sizes, averaged over 1000 runs. A: Time to 100% consensus. B: Time to 90% consensus. C: Time to 74% consensus. D: Time to 50% consensus.

spreading rate using level 1 managers as seeds versus level 2 managers on the Project network, while the Line-Org network shows a fairly flat advantage for each level increase in the system. This effect is also traceable to the structural differences in the layers of each network, as Table 1 and Fig 2 show that the Project network has an extremely dense central connection of nodes comprising largely of the first two management layers, while the Line-Org network is more evenly spread. The results in Fig 4 indicate that tapping into this dense core is of utmost importance to increased consensus as it allows spreading to occur throughout the elevated layers quickly and thus minimizing the difference between seeding among the management layers. Conversely, the tree-like layout of the Line-Org network emphasizes the more central position that each layer elevation grants and thus provides a flat benefit for each additional layer.

**SIR dynamics.** Extending to the SIR framework with a removal rate of $r = 0.2$, we perform a similar analysis as with the SI formulation. Fig 5 shows that the advantages of the Project network remain with the addition of the state $R$, providing a larger max adoption than the same dynamics on the Line-Org network. The only comparable value is with the bare minimum seed—a single staff node—and even there, unlike in the SI tests, the Project network slightly outperforms the Line-Org network. This change is due to the high likelihood of innovation death in the case of a single seed node in the SIR framework; in both networks a large amount of simulations will never spread past the seed node. This early death is even more likely in the Line-Org network since these isolated nodes have communities surrounding them to extinguish the innovation, while near-isolates in the Project network have only a single node with the ability to affect their state. In contrast, the strict SI framework does not allow for innovation death, and thus having the seed node have a higher number of neighbors simply speeds up the initial phase of the system coming to consensus. Further, outside of this fringe case the Project network provides significant advantages to the spreading process; at the top end, the Project network reaches over 70% of the total population while the Line-Org is unable to reach

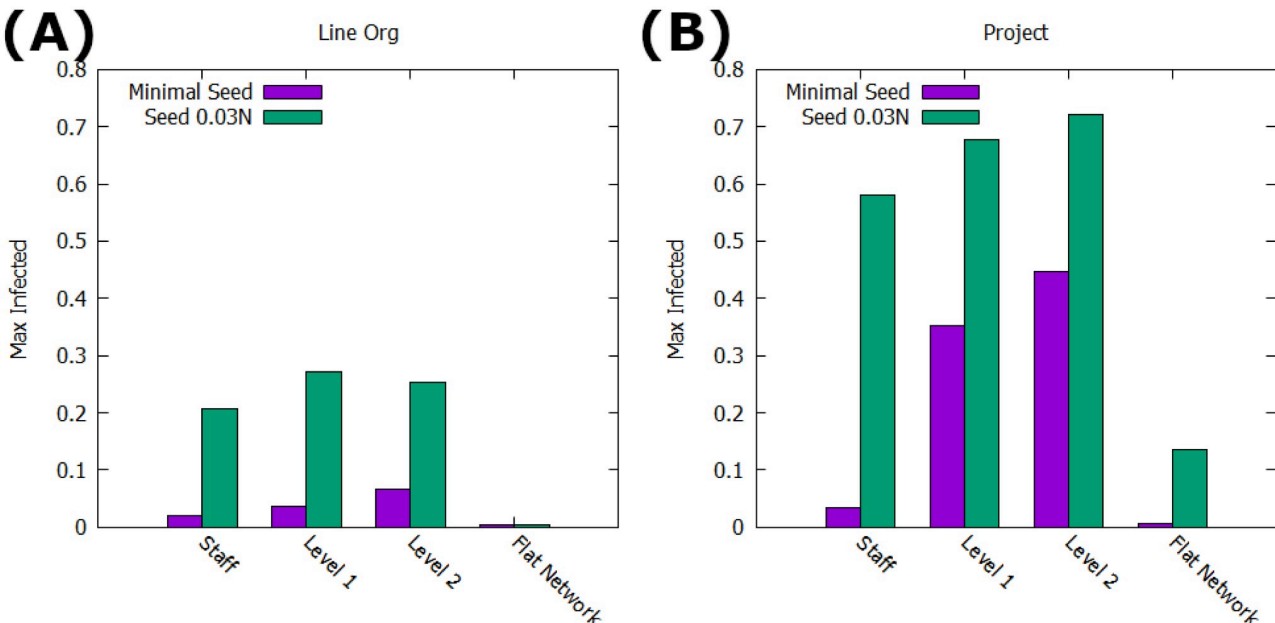

**Fig 5. Max adoption with SIR dynamics.** Shows the average maximum population of nodes in state $I$ for each combination of network, seed size, and seed level, averaged over 1000 runs. Includes a 'flat network' consideration, where management advantages are removed. A: Average max reach in the Line Org network structure. B: Average max reach in the Project network structure.

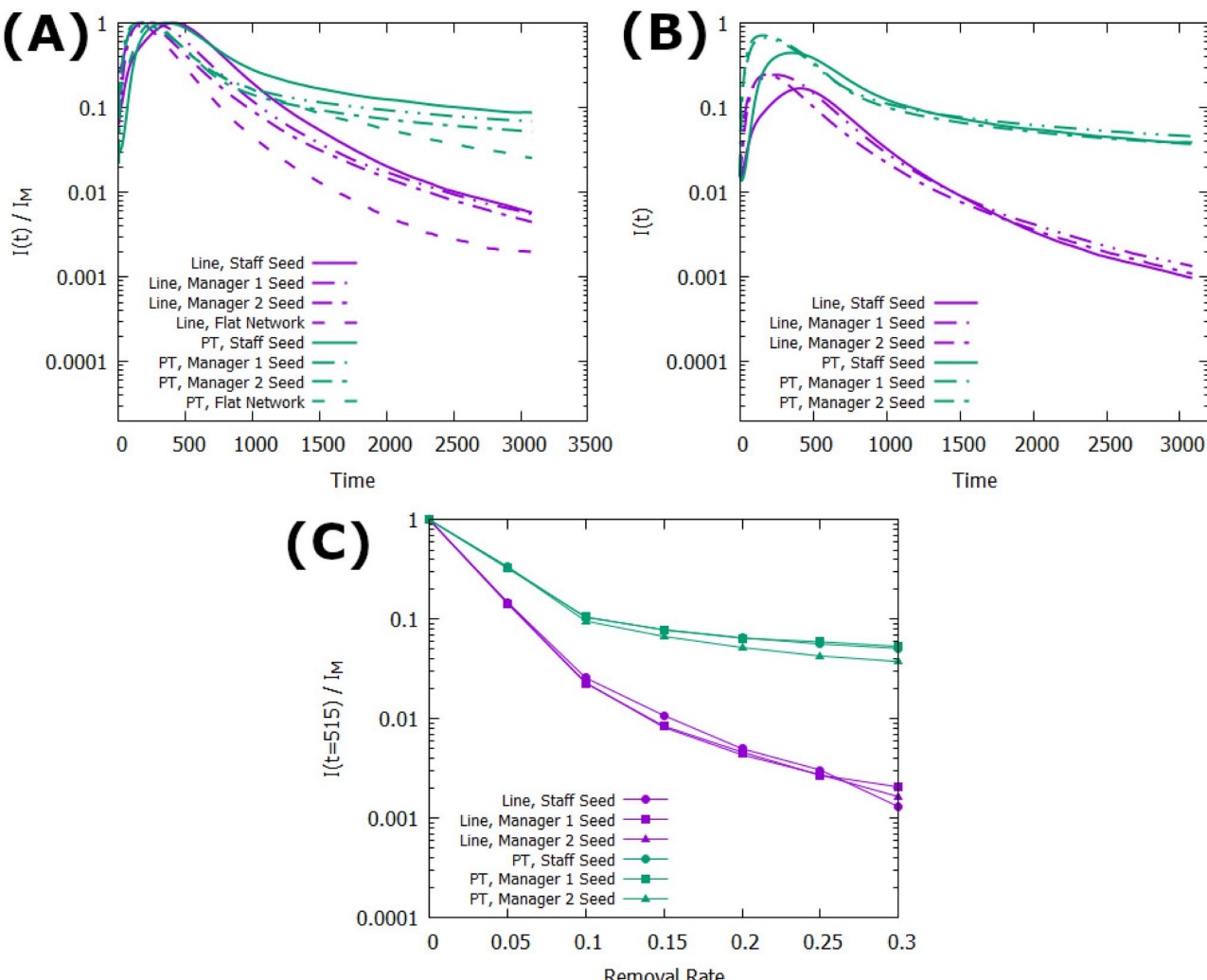

**Fig 6. Stability of _I_ population in long-time scenarios.** All seeds are size 0.015*N* and all data points are averaged over 1000 runs. A: The surviving percentage of the maximum infected population over time. B: The raw percentage of nodes in state *I* over time. C: Percent of the maximum adopted community remaining at time *t* = 515 with varying removal rates.

even 30% at its max. Further, Fig 5 also shows the benefit of the multilayered formulation leveraging the increased power of the higher layers, as the 'Flat Network' tests use the same edge requirements but remove the extra spreading rules for the management layers (effectively preserving the structure but 'flattening out' the network into a monoplex network). In this case, the maximum adoption in the system falls dramatically; for the Line-Org network, effectively no adoption occurs at all, and while the Project network has slightly more success it is still unable to produce a significant population of adopters without the multilayer support.

Finally, for innovation adoption understanding the longevity of the adopted population is just as important as the peak, leading us to consider the tail end of the adoption cycle and analyze not just the maximum number of individuals to reach state *I* but the long-time stability of this population. To this end, Fig 6 shows that the population of nodes in state *I* is considerably more stable in the Project network than the Line-Org network, leveling out much sooner and maintaining a long-time average of ∼ 10% of the maximum population versus the Line-Org's ∼ 1% without fully leveling off even out to 3500 epochs. Similarly, Fig 6(C) shows that

in addition to being more stable over time, the adoption process on the Project network is more resilient to higher removal rates in the system, showing minimal difference between removal rates above $r \sim 0.1$. On the Line-Org network, however, the system continues to be sensitive to increases in the removal rate up to much higher values of $r$. Additionally, Fig 6(A) shows that for both networks the multilayer structure also adds stability to the network, maintaining a higher infected percentage in the long time limit than the flat network counterparts and reaching this long-time limit sooner for both networks. Finally, the seeding population's effect on the long-time stability of the adopted populations should be noted, as while Fig 5 shows that the maximum population of adopted individuals increases when seeds are in the higher levels of the network, Fig 6(A) shows that seeding lower layers produces a more stable population in the long time limit. Even when accounting for the differences in maximum infection between the layers by looking at the raw percentage of nodes in state $I$ (shown in Fig 6(B)), the effect causes all layers to have similar long-time limit expectations, with a slight advantage towards seeding the second layer rather than the highest layer. Further, in the medium time regime, despite having a significantly lower maximum adoption, the staff layer seeding displays the largest infected population for a large amount of time due to its later peak and slower decay. This is likely due to the vulnerability of the initial adopters on the network; they are the first to spread the innovation and thus have a large impact on its reach, but they are also most at risk for permanent removal since there are extremely limited opportunities for reinforcement in the early time regime. As such, placing the seeds in the positions of the greatest reach allows for a greater initial impact, but limits the long time viability of the innovation due to the likelihood of losing such a key node in the process. Considering this, it may be more prudent to seed nodes *near* but not *in* positions of power, so that by the time the key nodes are infected there is more likely to be a small community to prevent their total loss from the population. This is supported by Fig 6(C), where the largest long-time population is the second layer in the system, where the optimal balance of initial impact without the loss of the most central nodes can be achieved.

**Seed selection strategies.** In the preceding sections we utilize a random seeding scheme to start the innovation spread in our simulations, a method that provides a good general view of how the layers perform when utilized in large sample sizes, but for total network reach is not an optimal seeding strategy. Additionally, there is a relationship between the demographics, connectedness, and power that an individual has within a system and their willingness to become an early adopter of an innovation [38, 39]. Thus, from both the perspectives of truly understanding the network potential and to capture this behavioral correlation, it is important to consider more sophisticated seeding strategies. To this end, we repeated the simulations above utilizing three different network centrality measures that are commonly considered for diffusion optimization [40]: node degree [41], VoteRank [42], and onion layer (an enhanced KCore number) [43]. For these tests, to reduce the randomness and allow for more direct comparison of the algorithms being tested, we coordinate execution utilizing a set of 1000 unique random seeds, one for each run, that allows for identical node initialization and attempted transmission pathways. Thus, for each run that the algorithms are applied to, the nodes in the comparison networks have the same thresholds and attempt to give the same dosages along the same pathways.

The results, in Fig 7, show how these seed selection strategies perform on the two networks. First, the Line-Org network remains much less conducive to spread than the Project network for all strategies. The overall conclusion that the more centralized co-working network is more conducive to spread holds, and even with more advanced selection strategies the Line-Org network never reaches a majority of the network adopted. Additionally, there are further indications suggesting that seeding near but not in high powered nodes can match or even beat the

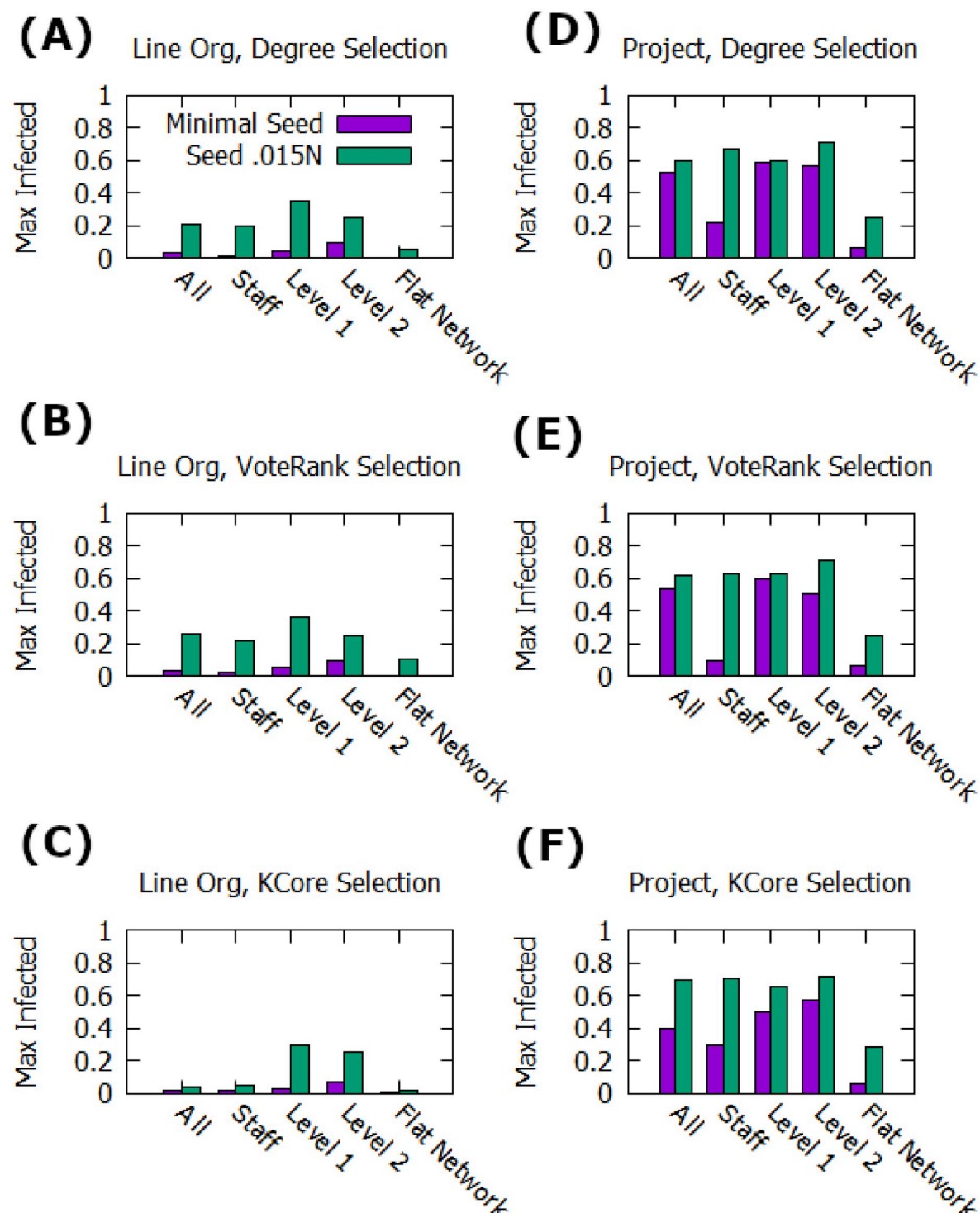

**Fig 7. Max adoption with SIR dynamics.** Shows the average maximum population of nodes in state *I* for each combination of network, seed size, and seed level, and selection algorithm averaged over 1000 runs. Includes a 'flat network' consideration, where management advantages are removed and an 'all' seed level where seeding is done by highest connected node regardless of level. A-C: Average max reach in the Line Org network structure for different seed selection strategies. D-F: Average max reach in the Project network structure for different seed selection strategies.

performance of seeding at the top of the network. When utilizing a non-minimal seed, for all three selection strategies seeding to Level 1 managers outperforms seeding to Level 2 managers on the Line-Org network and seeding staff outperforms seeding Level 1 managers on the Project network. The prior result is weakly present even in random selection, Fig 5(A), but due to the large number of poorly connected staff the latter result is only visible when using the strategies in Fig 7. Further, identifying well-connected individuals massively improves the performance with minimal seeds, nearly matching the performance of large seed sizes when applied at the manager levels. Finally, when comparing the flat network to one that retains the hierarchical scheme but seeds across all layers, the advantages of leveraging hierarchical spreading mechanics are stark. In every case, the average maximum adoption is more than doubled. While targeted management seeding can be nearly matched just by identifying nodes using centrality measures, leveraging their influence over downstream nodes changes the dynamics of the system greatly in favor of widespread adoption.

One of the keys to these hierarchical structures that allows them to respond so well to targeted seeding is the central node leadership of the working communities. As noted in [44], classical seed selection strategies often perform poorly on heavily community oriented networks and need a way to ensure the dispersal of seeds throughout the network. By targeting seeding at or near the management structure in these hierarchical networks, we are able to seed leader nodes with distinct communities, preventing issues of community overlap. Even when not actively targeting the managers, though, this phenomenon can be leveraged due to the manager skew in the network structure allowing leader nodes to be easily identified by the seed selection algorithms.

## Conclusion

In social science studies, isolation of groups and lack of institutional support is known to stifle spread and prevent new ideas from taking hold over current preferences [45, 46]. In practice, however, many institutional organizations show a high degree of clustering and hierarchy that creates a complex, multilayer structure for innovations to flow through. This makes the details of the best pathways and connections for optimal performance difficult to parse. Even when considered in a multiplex or multilayer format, prior studies have focused on random connections and simple spreading mechanics to lay the groundwork for how interconnection between layers can affect the overall dynamics. Here we extend these prior studies and present the case of a sharper focus on real-world systems based on large scale organizations and management hierarchies by building both synthetic and empirical hierarchical networks and running a variation of the Dosage model on them. In doing so we show that adding management layers to networks can have extreme effects on spreading processes, doubling the maximum reach of the social contagion on community-based RGG networks, as well as showing that while connections within the management layer can increase the spread, only a small level of connectivity between managers is sufficient to maximize this effect in most cases. In the case of empirical networks, we build two networks based on the same real-world dataset, one based on corporate defined administrative groups (the Line-Org network) and the other based on the working connections created by time spent in collaboration (the Project network). We show that the centralized properties of the Project network are far more conducive to innovation flow than the heavily clustered structure of the Line-Org network, and that the addition of multilayer management considerations is essential to having widespread innovation adoption on the network. Further, we show that contrary to common intuition, using the most highly influential nodes as the seed does not always provide the best long term outcomes. Highly influential nodes are occasionally outperformed by carefully selected nodes at lower levels, and

even when they do provide maximal peak adoption the long term stability of the adopted population is maximized by choosing nodes *near* the most influential nodes instead. This is a result of seed nodes being highly vulnerable to removal, which can permanently hinder the population of adopted nodes. In the networks used here, this means seeding the system with either staff or level 1 managers rather than the highest ranked nodes in the system.

These results advocate strongly for the increased effectiveness of innovation flow when both management structure and personal working relationships are leveraged rather than traditional methods of spreading via administrative lines, and suggest a more bottom-up rather than top-down approach for the adoptions of new innovations or policies. Although the analysis of the Polish Manufacturing dataset yielded similar patterns to the SNL dataset (see S1 Appendix), there are many areas that bear further investigation including alternative spreading processes or networks from other organizations. Additionally, the network frameworks here separate out the administrative and working networks into two distinct structures, but one of the strengths of multilayer networks is the ability to handle different types of communication modes between nodes. Here we consider the case of two different inter-connected networks with differing intra-layer connections, but they could be combined into a single more complex structure with both the interconnected network layers of the management hierarchy and the multiplex communication channels within each layer. Finally, the dosage model used here has a strict and individual form of memory for the dosage accumulation, but more complex and cognitively informed model memory handling for individuals in the presence of hierarchical power dynamics similar to other recent efforts in the area would be another possible avenue for future work [47]. Still, the work presented here yields another step towards the full picture of real-world spreading, where communication channels are not always clean and distinct such as in the case of large-scale organizations adopting new innovations.

## Supporting information

**S1 Appendix. Background plots, elaborations, and Polish manufacturing data.** Analysis of the hierarchical spreading model on a Polish Manufacturing network as a secondary empirical source to supplement the presented SNL data.
(ZIP)

**S1 Sample code. Synthetic network creation python script.** Sample code for generating the hierarchical synthetic networks used in this paper. The code is written in Python and uses the NetworkX package for the creation of established network structures and community detection implementation [48].
(ZIP)

**S1 Fig.**
(TIF)

**S2 Fig.**
(TIF)

**S3 Fig.**
(TIF)

**S4 Fig.**
(TIF)

**S5 Fig.**
(TIF)

**S1 File. SNL node list.** This file contains a list of node ID's and network levels for all nodes used in the Line-Org and Project networks.
(CSV)

**S2 File. SNL Line-Org edge list.** This file contains a list of all edges between the nodes (contained in S1 File) for the Line-Org network.
(CSV)

**S3 File. SNL Project edge list.** This file contains a list of all edges between the nodes (contained in S2 File) for the Project network.
(CSV)

**S4 File. Polish Manufacturing node list.** This file contains a list of node ID's and network levels for all nodes used in the Line-Org and Project networks.
(CSV)

**S5 File. Polish Manufacturing Line-Org edge list.** This file contains a list of all edges between the nodes (contained in S5 File) for the Line-Org network.
(CSV)

**S6 File. Polish Manufacturing emails.** This file contains a list of all edges between the nodes and the number of emails (contained in S5 File) for the Email network.
(CSV)

## Author Contributions

**Conceptualization:** Casey Doyle, Thushara Gunda, Asmeret Naugle.

**Data curation:** Casey Doyle, Thushara Gunda.

**Formal analysis:** Casey Doyle, Thushara Gunda.

**Funding acquisition:** Asmeret Naugle.

**Investigation:** Casey Doyle, Asmeret Naugle.

**Methodology:** Casey Doyle, Thushara Gunda.

**Project administration:** Asmeret Naugle.

**Visualization:** Casey Doyle.

**Writing – original draft:** Casey Doyle.

**Writing – review & editing:** Casey Doyle, Thushara Gunda, Asmeret Naugle.

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
