## [Decision Letter · Decision Letter 0]

25 Jun 2020

PONE-D-20-06635

Hierarchical effects facilitate spreading processes on synthetic and empirical multi-layer networks

PLOS ONE

Dear Dr. Doyle,

First of all, we would like to thank you and your coauthors for submitting your manuscript to PLOS ONE and for the patience during the review process. We apologise for the huge delay we incurred in taking a decision, mainly caused by the current pandemic situation and that one of the two secured reviewers unfortunately had to drop out from the process without returning a report.

From a critical reading of the comments from the Reviewer #1, we conclude that your work should represent an interesting addition to the current literature devoted to spreading processes in networks. At the same time, according to the referee's comments, there are serious concerns about how the research has been performed. In particular, the referee points out problems with definitions and description of the model, and several issues about the different conducted experiments making difficult to support the  conclusions drawn in the manuscript.

On this basis, we regret that we cannot make you an offer of publication. However, we would be willing to consider a rebuttal and a revised version. At the same time, we imagine that carrying out some of the revisions may require a significant amount of additional work on your part.

Should you consider you can overcome all the referee's criticisms, and you wish your manuscript to be reconsidered in PLOS ONE, please respond point-by-point to all of the referee's comments and revise your manuscript as appropriate.

We look forward to receiving your revised manuscript.

Kind regards,

Irene Sendiña-Nadal

Academic Editor

PLOS ONE

Journal Requirements:

2.Thank you for clarifying how the empirical networks data was accessed. Please add the following text to your Methods: Empirical Networks section: 'Access to the data was overseen by the Sandia National Laboratories data governance committee, and all data was fully anonymized before being made available to the authors.

3.Thank you for stating the following in the Financial Disclosure section:

[ This study was funded by Sandia National Laboratories, a multimission laboratory managed and operated by National Technology & Engineering Solutions of Sandia, LLC, a wholly owned subsidiary of Honeywell International Inc., for the U.S. Department of Energy’s National Nuclear Security Administration under contract DE-NA0003525.

The Sandia National Laboratories website is https://www.sandia.gov/

The funders helped procure the data for this work.]

We note that you received funding from a commercial source: [National Technology & Engineering Solutions of Sandia, LLC]

Reviewers' comments:

Reviewer's Responses to Questions

**Comments to the Author**

1. Is the manuscript technically sound, and do the data support the conclusions?

Reviewer #1: Partly

2. Has the statistical analysis been performed appropriately and rigorously? 

Reviewer #1: No

3. Have the authors made all data underlying the findings in their manuscript fully available?

Reviewer #1: Yes

4. Is the manuscript presented in an intelligible fashion and written in standard English?

Reviewer #1: No

5. Review Comments to the Author

Reviewer #1: This paper presentments quite exciting topic which is very important in nowadays network science. However, there are still some issues to address before the paper can be published.

Major issues

1. It seems that through the paper authors use information, innovation and influence spread as synonyms when, in fact, they are three different sociological phenomena. Thus, please clearly state what is being modeled and evaluate; spread of information or spread of innovation.

2. Having 1, please elaborate on how the selected model (SIR) works in that scenario and how real processes can be translated to mechanisms of the model. For example, if we have spread of information we would have S – do not have information, I – informed, R – forget the information (then maybe it will be better to consider SIS). Innovation case -> S – not adopted, I – adopted, R – forgot/rejected the innovation -> but why one cannot adopt it again? Moreover, if we can reject the innovation why we cannot go straight from I to R based on many neighbouring nodes which already rejected the innovation? Please fully justify the model and the reasoning behind each element of the model.

3. Having 1 and 2, please clearly justify selected parameters for example, why a network is clustered into 60 clusters (is it always 60 or depends on the network size?, why 60?)? Why the probability of transition to R is 0.2, why not 0.1 or 0.5, what will be the effect on the process depending on selected probability? Why 370hours of collaboration and not 100 or 500? etc.

4. Selected networks are all of the same size 1000-1800 nodes which do not allow for results generalization. Please use both smaller and bigger networks to address both SMEs and large corporations at least for simulated data. For real networks, please use other “company/institution” datasets like Manufacturing emails (Radoslaw Michalski, Sebastian Palus, and Przemyslaw Kazienko. Matching organizational structure and social network extracted from email communication. In Lecture Notes in Business Information Processing, volume 87, pages 197--206. Springer Berlin Heidelberg, 2011.), Enron emails and Enron structure (Palus, S., Brodka, P., & Kazienko, P. (2011). Evaluation of organization structure based on email interactions. International Journal of Knowledge Society Research (IJKSR), 2(1), 1-13.) and others.

5. Selected approach for seed selection does not allow for (1) easy comparison of results between networks of various sizes and (2) results reproducibility. For (1), please consider using a percentage of a network as seeds (e.g. 1% or 0.5% of nodes will be seeds) instead of a fixed number of seeds. For (2), please consider using some seed selection strategy instead of random selection, e.g. top % of nodes with the highest degree. There is many seed selection strategies for multilayer and single layer networks (Erlandsson, F., Bródka, P., & Borg, A. (2017, November). Seed selection for information cascade in multilayer networks. In International Conference on Complex Networks and their Applications (pp. 426-436). Springer, Cham.). The most useful in case of the problem presented in the paper and high cauterization of nodes might be community structure based method like (He, J.-L., Fu, Y. & Chen, D.-B. A novel top-k strategy for influence maximization in complex networks with community structure. PloS one 10, e0145283 (2015).). Please also use simple degree as a benchmark. Please consider using some early adopters mechanism to introduce the real appearance of innovation in the company, which is usually done by people with some key characteristic (leaders, influencers, innovators) rather than random people.

6. What was the dosage threshold for each node? Was it always the same or it was selected randomly in each run. If randomly each time how you compare results between two networks if we have random seeds, random thresholds and random transition to R each time? Please consider coordinated execution approach similar to one used in (Jankowski, J., Szymanski, B. K., Kazienko, P., Michalski, R., & Bródka, P. (2018). Probing limits of information spread with sequential seeding. Scientific reports, 8(1), 1-9.) where you generate for example 10 000 instances of nodes threshold and R transitions plus add the seeds selected according to some heuristic/strategy. Then we can compare the results between networks for each instance and see what affect the network topology and/or seed selection strategy have on the spreading process.

7. Please evaluate if results are statistically significant

Minor issues

8. What is the difference between Dosage SI model and Linear threshold model –on the first glance, it looks the same. If they are the same, please consider noting that in the manuscript.

9. Please unify the naming convention in the paper, e.g. sometimes it is multilayered, sometimes multi-layered

6. PLOS authors have the option to publish the peer review history of their article (what does this mean?). If published, this will include your full peer review and any attached files.

Reviewer #1: No

---

## [Author Response · Author response to Decision Letter 0]

22 Dec 2020

In depth responses to each raised point by the reviewer and editor is included in the Response to Reviewers letter.

Editor:

1. We have updated the the file naming convention as well as the referencing inside the manuscript to conform with requirements.

2. The requested statement was added to the manuscript.

3. The cover letter has been updated to comply with the request.

Reviewer:

1. A full explanation has been added to the response letter, as well as updates throughout the manuscript to clarify the point.

2. This too has been included in the manuscript and the response letter to clarify and focus the paper.

3. Justification for parameter choices have been added to both the response and the manuscript.

4. We have run new simulations and conducted new analysis to comply with this request. This includes a new figure in the main body of the manuscript as well as the new S1 Appendix file that considers a new empirical dataset.

5&6. The requested analysis has been completed to confirm the reproducibility of the results, leading to a new section in the manuscript.

7. Explanation of the statistical significance of the Monte Carlo simulations is included in the response letter.

8. A clarification on the differences between the models is included in the response letter.

9. The manuscript has been cleared of these inconsistencies.

---

## [Decision Letter · Decision Letter 1]

19 Jan 2021

PONE-D-20-06635R1

Hierarchical effects facilitate spreading processes on synthetic and empirical multi-layer networks

PLOS ONE

Dear Dr. Doyle,

Thank you for submitting your manuscript to PLOS ONE. After careful consideration, we feel that it has merit but does not fully meet PLOS ONE’s publication criteria as it currently stands. Therefore, we invite you to submit a revised version of the manuscript that addresses the points raised during the review process.

We look forward to receiving your revised manuscript.

Kind regards,

Irene Sendiña-Nadal

Academic Editor

PLOS ONE

Reviewers' comments:

Reviewer's Responses to Questions

**Comments to the Author**

1. If the authors have adequately addressed your comments raised in a previous round of review and you feel that this manuscript is now acceptable for publication, you may indicate that here to bypass the “Comments to the Author” section, enter your conflict of interest statement in the “Confidential to Editor” section, and submit your "Accept" recommendation.

Reviewer #1: (No Response)

2. Is the manuscript technically sound, and do the data support the conclusions?

Reviewer #1: Yes

3. Has the statistical analysis been performed appropriately and rigorously? 

Reviewer #1: N/A

4. Have the authors made all data underlying the findings in their manuscript fully available?

Reviewer #1: No

5. Is the manuscript presented in an intelligible fashion and written in standard English?

Reviewer #1: Yes

6. Review Comments to the Author

Reviewer #1: The paper has been significantly improved, addressing most of the issues.

I am satisfied with the replay and changes made regarding comment 1, 4, 5, 6, 8 and 9. However, there are still some issues related to comment 2 and 3 in the manuscript

2. The model description is still not detailed enough with regards to motivation and reasons behind using SIR to spread innovations e.g.

• Can the innovation be ever accepted or at the end, it will always be rejected by the population? How this relates to real-world cases? Maybe fig 1 could extend (by adding 1b) by adding a theoretical S I R states dynamics – something like this https://en.wikipedia.org/wiki/Compartmental_models_in_epidemiology#/media/File:Graph_SIR_model_without_vital_dynamics.svg or this https://en.wikipedia.org/wiki/Compartmental_models_in_epidemiology#/media/File:SIR_trajectory.png and discuss this, also extend fig 6 could be extended in the same way, by adding the state dynamics from simulations

• According to line 132-133, “there is no state transition directly from I to R under the assumption of good faith on the part of the agents”. Should not it be from S to R since in line 91-93 we have “Conversely, if they are in state I and Dti < di, they have a probability r = 1=T, or r = 0:2, of entering the removed state (R), signifying that they have abandoned the new innovation due to perceived lack of support.”?

3a. Since the c=.06N, please provide the number of partitions in each simulation case. What is the distribution of partitions sizes? In line 150, please change Nstaff to N or introduce what Nstaff means.

3b. “Then, each community is given a `manager' node that is placed on a second layer in the graph and attached only to the downstream nodes that are within its assigned community.” How is it done? Randomly? How does it look in a real network? Does the “more important manager” (e.g. with a higher degree or some other user importance measure in manager layer) manage bigger groups?

3c Since some parameters are linked, e.g. removal rate and memory window I would like to know if other combinations of parameters have been tested and what would be the memory window if removal rate would be 0.1 or 0.3? Please include this discussion in the paper.

Additionally, since many recent papers (e.g. Social Networks through the Prism of Cognition, Complexity, vol. 2021, Article ID 4963903) indicates that people tend to forget, I would like to know if you have considered adding forgetting functions to your model, i.e. that older dosages have a weaker effect on the person?

Minor new comments

10 Please add a legend to fig 2 or add colours to the caption

11 Please unify the Y-ax on figures 3a and 3b (scale); 6a with 6b and 6c (formating);

12 Please make all data and code needed for reproduction available, including all synthetic networks generated for simulations which are currently unavailable. You can use CodeOcean or something similar to make your experimental environment capsule.

7. PLOS authors have the option to publish the peer review history of their article (what does this mean?). If published, this will include your full peer review and any attached files.

Reviewer #1: No

---

## [Author Response · Author response to Decision Letter 1]

5 Mar 2021

Reviewer comments:

2. We have thoroughly discussed the theoretical aspects of this system both in the response to reviewers letter and have added a summary of this response into the paper. This includes the real-world analogue of our system, the long time behavior, and the timeseries state curves requested. Additionally, we have fixed the critical typo pointed out by the reviewer.

3. We have changed and clarified the description of the synthetic network build process as well as provided sample code in order to make this more accessible and reproducible for readers. We have also included an in depth discussion of the community size distribution produced by our method, including a fit for the PDF. Finally, we have furthered the theoretical discussion of the effect of different parameter values and combinations, highlighting prior work in this area that considers this on an analytical level for general values of the parameters of interest, and have discussed how these parameters relate to the phenomenon of forgetting in social systems.

10. This figure has been updated with a legend.

11. These figures have been reproduced to be more stylistically consistent.

12. We have included further discussion on the creation process of these structures as well as added S2 Sample Code, a python script that produces the synthetic network structures used in the paper.

---

## [Decision Letter · Decision Letter 2]

6 Apr 2021

PONE-D-20-06635R2

Hierarchical effects facilitate spreading processes on synthetic and empirical multi-layer networks

PLOS ONE

Dear Dr. Doyle,

Thank you for submitting your manuscript to PLOS ONE. According to the referee's suggestion, it is convenient to add the material produced during the second round of revision into the supplementary information as it could better help readers to follow the work. Therefore, we invite you to submit a revised version of the supplementary information and the manuscript where needed to make appropriate reference to the supplemental material. 

We look forward to receiving your revised manuscript.

Kind regards,

Irene Sendiña-Nadal

Academic Editor

PLOS ONE

Journal Requirements:

Reviewers' comments:

Reviewer's Responses to Questions

**Comments to the Author**

1. If the authors have adequately addressed your comments raised in a previous round of review and you feel that this manuscript is now acceptable for publication, you may indicate that here to bypass the “Comments to the Author” section, enter your conflict of interest statement in the “Confidential to Editor” section, and submit your "Accept" recommendation.

Reviewer #1: All comments have been addressed

2. Is the manuscript technically sound, and do the data support the conclusions?

Reviewer #1: Yes

3. Has the statistical analysis been performed appropriately and rigorously? 

Reviewer #1: N/A

4. Have the authors made all data underlying the findings in their manuscript fully available?

Reviewer #1: Yes

5. Is the manuscript presented in an intelligible fashion and written in standard English?

Reviewer #1: Yes

6. Review Comments to the Author

Reviewer #1: The paper has been further improved and in my opinion is ready for publication; however, I would like the authors to include the materials from their round 2 replay (those not included in the main paper) as a supplementary material (they can extend S1 or add a new supplement), since I think some of them might help the reader to faster understand some elements, for example one look at fig 1 and 2 from the replay explains a lot.

7. PLOS authors have the option to publish the peer review history of their article (what does this mean?). If published, this will include your full peer review and any attached files.

Reviewer #1: No

---

## [Author Response · Author response to Decision Letter 2]

10 May 2021

The requested material from the reviewer response has been added to the supplementary information as an extension to the S1 Appendix, and both the text explanations and figures referenced have been included. We thank the reviewer for their time and patience with this submission.

---

## [Editor Report · Decision Letter 3]

14 May 2021

Hierarchical effects facilitate spreading processes on synthetic and empirical multi-layer networks

PONE-D-20-06635R3

Dear Dr. Doyle,

We’re pleased to inform you that your manuscript has been judged scientifically suitable for publication and will be formally accepted for publication once it meets all outstanding technical requirements.

Kind regards,

Irene Sendiña-Nadal

Academic Editor

PLOS ONE
---

## [Editor Report · Acceptance letter]

27 May 2021

PONE-D-20-06635R3 

Hierarchical effects facilitate spreading processes on synthetic and empirical multilayer networks 

Dear Dr. Doyle:

I'm pleased to inform you that your manuscript has been deemed suitable for publication in PLOS ONE. Congratulations! Your manuscript is now with our production department. 

Kind regards, 

on behalf of

Dr. Irene Sendiña-Nadal 

Academic Editor

PLOS ONE